# Spatial-Temporal Differentiation and Spatial Interaction Effect Analysis of Residents' Consumption Capacity and Consumption Upgrading in China

**Xiaowei Xing** [1,*] and **Azhong Ye** [2]

1   School of Business Administration, Northeastern University, Shenyang 110167, China
2   School of Economics and Management, Fuzhou University, Fuzhou 350108, China
*   Correspondence: 2010444@stu.neu.edu.cn

**Abstract:** To investigate Chinese residents' consumption capacity and consumption upgrading, this study uses the Moran index, the Dagum Gini coefficient method, and the SPVAR model to analyze the spatial-temporal differentiation and interactions between the two; the entropy method is employed to measure the development level of both factors. As per the results, from 2008 to 2020, the levels of consumption capacity and consumption upgrading have increased rapidly and are distinctly unbalanced, but there is a clear trend of convergence between the developments in each province, and both show spatial agglomeration, mainly in the "high–high" and "low–low" clusters. The "high–high" clusters are primarily concentrated in the more economically developed eastern coastal and central fast-rising regions, with inter-regional differences being the main reason for the regional differences between the two. In terms of spatial interaction, the two are Granger causalities, with positive feedback effects that promote each other, and have a "center–periphery" spatial spillover effect, where the center generates spatial spillover and also suffers reverse shocks from the periphery, while the shock effect is limited by inter-provincial economic ties and geographical distance.

**Keywords:** consumption capacity; consumption upgrading; spatial-temporal differentiation; spatial interaction; Dagum Gini coefficient

## 1. Introduction

As the starting and ending point of the economic cycle, consumption is a crucial factor that embodies the final demand of society. It plays a fundamental pulling and guiding role in economic development and is a realistic choice for smoothing the double cycle, improving the quality of residents' lives, and achieving common prosperity. The trend of consumption upgrading and the acceleration of changes in social lifestyles have led to the need to guide enterprises to upgrade their production structure, drive the transformation of industrial structure, optimize the structure of economic development, promote the transformation of economic growth momentum, and realize the upgrading of economic quality and efficiency [1–3]. Therefore, promoting high-quality economic development through consumption upgrading has become an important macro policy direction. However, the sustained and stable release of consumption potential, the acceleration of consumption upgrading, and the construction of a long-term mechanism for consumption-driven economic growth cannot be achieved without a solid foundation of residents' consumption capacity [4]. As consumption subjects, residents can directly influence the final demand of society. Only when they are able to consume, dare to consume, and are willing to consume can they increase the quantity of consumption and expand consumption demand, giving rise to new consumption and business models, and accelerating the process of social reproduction. This will help improve the structural imbalance between demand and supply, change the income and wealth levels of the population, and achieve a jump in consumption

capacity. Therefore, it is crucial to prioritize the enhancement of residents' consumption capacity to promote high-quality economic development through consumption upgrading.

Overall, China's consumption has become a crucial driver of high-quality economic development, as evidenced by a declining Engel coefficient, increasing service consumption, and final consumption contributing to over 60% of the GDP. This suggests that consumption is optimized and upgraded as consumption capacity grows. However, due to future uncertainties, consumers must allocate more funds towards "aftercare" such as healthcare, housing, education, and retirement. It is indisputable that despite having money, residents are hesitant to consume [4–7]. Moreover, the growth rate of residents' consumption levels is slowing down, with per capita consumption expenditure and total retail sales of social consumption growing only 5.5% and 8%, respectively, in 2019, the lowest in the past two decades. The rise in sales of low-level goods as well as cheap and shoddy goods and services on shopping sites [1] also suggests that the consumption situation is less optimistic. Consequently, the discussion on "upgrading–downgrading consumption" has become a hot topic in academia and industry.

To establish a new "double-cycle" development pattern, enhancing the consumption capacity and promoting consumption expansion are crucial for sustainable and reliable economic growth. However, China's vast population results in significant regional differences in consumption culture, economic development, and resource endowment, leading to disparities in residents' consumption capacities and upgrading levels across regions [2,3,8,9]. Thus, studying consumption capacity and upgrading is essential for formulating effective macroeconomic policies that synergistically target consumption potential and promote regional economic development. Higher levels of consumption capacity and upgrading indicate stronger consumption capacity, social demand, and the ability to drive industry prosperity. However, measuring regional differences and the spatial dependence of consumption capacity and structural upgrading, as well as analyzing their dynamic interaction under regional economic integration, is crucial for exploring regional synergy in practicing "domestic demand-led" economic growth, enhancing residents' happiness, and achieving common prosperity.

## 2. Literature Review

With the slowdown of China's economic growth, the sustainability of investment and export drivers among the "troika" of economic growth is gradually weakening, and the structural contradiction between investment, export, and consumption is becoming increasingly prominent [7]. Consequently, scholars have begun to focus on consumption upgrading and enhancing residents' consumption capacity. Scholars at home and abroad have carried out a large number of analyses on the influencing factors, impact effects, evaluation indicators, and spatial differentiation of consumption upgrading and consumption capacity.

In terms of influencing factors, consumption theories unanimously agree that income is the most crucial determinant of consumption. Empirical analyses substantiated that China's low-income population is the primary reason for its lack of consumption and weak "consumption capacity." However, with the inclusion of uncertainty in consumption research, many scholars have found that uncertainty about the future of housing, retirement, healthcare, and children's education is also an important factor in dampening residents' willingness to consume and forcing them to increase precautionary savings [5,7,10]; however, this is not conducive to the expansion and sustainable upgrading of consumption. In addition to being influenced by consumers' own factors such as income and expectations, consumption expansion and upgrading are also influenced by cultural and psychological factors such as social, cultural, and psychological perceptions related to consumption habits [6,8,11,12], technological progress, new business models, supply side changes such as consumer regimes [1,13,14], and shocks such as income disparity and consumption stratification due to uneven economic and social development [15–17]. Therefore, China's central government and localities have been doing everything possible to stabilize and broaden employment, adjust national income patterns, improve social security, create a

harmonious environment for consumption, and continuously increase the proportion of the income of middle- and low-income residents, in order to increase residents' income and raise their willingness to consume, so that they are willing, able, and dare to consume, and steadily improve their consumption capacity [4,6,7].

In terms of impact effects, theoretical analysis shows that the increase in consumption capacity can promote the upgrading and optimization of consumption, especially of product structure, prompting the transformation of the leading industries of the economy from traditional to modern service industries and accelerating economic transformation and upgrading [3,8,18]. The new consumption and industries emerging in the process of consumption upgrading will accelerate the reselection of the industrial chain division of the labor system, changing the scale and structure of investment in the three industrial structures accordingly and affecting industrial development and economic growth [13,14,19]. The steady development of industries and the economy is not only a sufficient condition for achieving high-quality development of China's economy but also helps stabilize employment, protect people's livelihoods, prevent risks, stimulate the vitality of market subjects, continuously improve people's quality of life, and enhance residents' consumption capacity [20–22]. This strengthens the positive feedback of coupling and synergy between residents' consumption capacity and the consumption upgrading effect. However, existing studies have mainly explored the effect of increased consumption capacity on consumption upgrading from the consumer power perspective [23,24], and there is little literature on the comprehensive analysis of the dynamic interaction between consumption capacity and consumption upgrading from the perspective of spatial spillover.

Various methods are used to measure consumption upgrading and consumption capacity, such as the "Engel coefficient", the "new Engel coefficient", the "development coefficient", the share of different types of consumption expenditure, and the elasticity of demand or expenditure for different types of consumption goods. These methods, while simple and easy to apply, have limitations in reflecting the overall role of consumption subjects, the consumption environment, and social security. Therefore, adopting a comprehensive indicator evaluation method can provide a more comprehensive reflection of consumption upgrading and consumption capacity. For example, Liang [25] measures household consumption capacity from the perspective of individuals' and households' subjective evaluation of their consumption capacity, using low-income households' evaluation of whether their household economic situation can sustain various levels of consumption. Ye [8] proposed that consumption upgrading should include five levels of content: economic development and structural upgrading, consumption content upgrading, consumption capacity upgrading, consumption pattern upgrading, and consumption environment upgrading. However, owing to the differences in the starting points and purposes of the studies, there are inconsistencies in the construction of the comprehensive evaluation index system, leading to large differences in the results of measuring consumption capacity and consumption upgrading. These differences can affect the subsequent acquisition of relevant scientific research findings.

In terms of spatial divergence analysis, the extant literature on regional differences in consumption upgrading and capacity and the dynamic evolution of their distribution can be categorized into two perspectives. The first perspective investigates the factors that influence regional differences in consumption upgrading and capabilities. For instance, Zhang et al. [2] and Hong et al. [9] employed spatial econometric models to explore the factors affecting inter-provincial differences in consumption upgrading and consumption capacity in China. Ye [8] examined the factors influencing the evolutionary trend of inter-provincial differences in the level of consumption upgrading in China at the level of absolute beta convergence. The second perspective relies on a simple comparative and correlation analysis of consumption upgrading and capacity in China. Research in this area mainly focuses on data comparison and the analysis of differences in China's consumption upgrading and capacity by region, class, and area type (rural or urban) based on the measurement results [3,18,25]. Some scholars have verified the spatial correlation

test and coefficient of variation of China's consumption upgrading or the phenomenon of spatial divergence in the development of consumption capacity [9,26]. However, only a few scholars, such as Ye [8], have used a distributional dynamics model to conduct a preliminary analysis of the dynamic evolution of the distribution of consumption upgrading in China. Moreover, only a few scholars have used the Dagum Gini coefficient and its decomposition method to conduct an in-depth investigation of regional differences in resident consumption capacity.

In summary, there is a paucity of studies on the decomposition of regional differences in consumption upgrading and consumption capabilities and their spatial interactions. Based on this, to objectively and effectively identify the characteristics of the uneven development of consumption capacity and consumption upgrading, as well as the spatial interaction effects, this study uses the spatial Moran index, the Dagum Gini coefficient, and the spatial vector autoregressive model (SPVAR) to conduct an empirical study; consequently, it provides a reference for China to form a synergy between regions to expand domestic demand. The possible innovations of this study are as follows. First, most of the current literature on consumption capacity and consumption upgrading adopts a single indicator approach, ignoring the fact that consumption activity is a more complex process. To overcome this shortcoming, we re-measured consumption capacity and consumption upgrading by constructing a system of evaluation indicators which is conducive to the extension and enrichment of the theoretical framework of consumption. Second, few scholars have used the Moran index and Dagum Gini coefficient to investigate spatial differences in the distribution of consumption capacity and consumption upgrading as correlation analysis can better reflect the real level and dynamic distribution of consumption capacity and consumption upgrading in each region. Third, unlike other studies on the synergy of variables, this study does not focus on the level of coupling and synergistic development of residents' consumption ability and consumption upgrading but uses the SPVAR model to further explore the dynamic spatial interaction between residents' consumption capacity and consumption upgrading from the perspective of regional spatial correlation—the biggest difference between this study and previous studies.

## 3. Research Design

### 3.1. Evaluation of Consumption Upgrading and Capacity

3.1.1. Indicators for Evaluating Consumption Capacity

Currently, there is limited research on the comprehensive evaluation of consumption capacity, with most studies focusing on objective factors, such as income or expenditure. However, scholars such as Yu et al. [3], Liu [4], Liu and Mi [18] emphasized the importance of subjective factors, including consumers' willingness to buy and the social environment, in shaping consumption behavior. The 14th Five-Year Plan for National Economic and Social Development of the People's Republic of China and Vision 2035 also highlight the need to improve quality of life and increase income to expand the middle-income group and unleash consumption potential. Premier Li Keqiang further emphasized the importance of increasing residents' income, enhancing their consumption capacity, and improving the consumption environment by considering various elements such as income, employment, security, environment, and economic development. Table 1 presents a system of indicators for evaluating people's consumption capacity (CSP), which encompasses 23 indicators across four criteria levels: income, social security, consumption environment, and willingness to consume.

**Table 1.** Comprehensive indicator system of consumption capacity.

| Guideline Level | Specific Indicators | Indicator Description/Measurement | Indicator Attributes | Indicator Weights |
|---|---|---|---|---|
| Resident income | Income | Disposable income per capita | + | 0.057 |
| | Income growth rate | Growth rate of disposable income per capita | + | 0.012 |
| | Wage income | Average wage of employed persons in urban units | + | 0.054 |
| Social Security | Registered urban unemployment rate | - | − | 0.032 |
| | Social insurance participation rate | (Number of participants in basic urban pension insurance at the end of the year + number of participants in unemployment insurance at the end of the year)/number of people at the end of the year | + | 0.05 |
| | Number of health technicians per 10,000 people | - | + | 0.03 |
| | Number of beds in medical institutions per 10,000 population | - | + | 0.028 |
| | Social security situation | Local financial social security and employment expenditure | + | 0.066 |
| | Job security | Number of corporate units | + | 0.117 |
| Consumer environment | Internet Development | Length of long distance fiber optic cable routes | + | 0.102 |
| | Postal industry business network radiation level | Number of postal outlets/land area | + | 0.012 |
| | Road network density | (road mileage + rail operating mileage)/area | + | 0.05 |
| | Cargo turnover | - | + | 0.103 |
| | Passenger Turnover | - | + | 0.068 |
| | Urban public transport vehicles | Public transport vehicles per 10,000 people in cities | + | 0.032 |
| | Level of financial development | Financial sector value added/GDP | + | 0.043 |
| | Inflation rate | Consumer Price Index | − | 0.007 |
| Willingness to consume | Years of schooling per capita | (Number of primary school population × 6 + number of junior high school population × 9 + number of senior high school population × 12 + number of tertiary and above population × 16)/population, all data are from population sample | + | 0.008 |
| | Percentage of population in tertiary education | Percentage of population aged 6 years and over in tertiary education | + | 0.043 |
| | Development of higher education | Average number of students enrolled in higher education per 100,000 population | + | 0.035 |
| | Illiteracy rate | Percentage of population aged 15 and over who are illiterate | − | 0.006 |
| | Urban–rural income gap | Ratio of disposable income per capita between urban and rural residents | − | 0.027 |
| | Total dependency ratio | - | − | 0.008 |

### 3.1.2. Indicators System for Consumption Upgrading

There have been numerous discussions both within and outside the academic community regarding the definition of consumption upgrading. However, the fundamental idea behind it remains the same—consumption upgrading entails the improvement of both the "quality" and "quantity" of consumption. This implies not only a shift from material

to service-based (spiritual) consumption or from fulfilling basic needs to cultivating enjoyment needs but also involves the coordination of different consumer goods, the total amount of consumption, and the consumer's consumption capacity. Furthermore, it directly reflects the upgrading of consumption patterns and changes in consumer attitudes during consumption [27]. Consumption upgrading relies on the overall association between all elements within the system and its subsystems. Therefore, obtaining biased conclusions is inevitable if consumption upgrading is solely measured from the perspective of the content of the consumption object while ignoring the role of consumption patterns and concepts of the consumption subject. This study draws on the definition of consumption upgrading and the scope of content covered by scholars such as Du [27] and Ye [8] to establish a consumption upgrading index. The index includes total social consumption, residents' consumption levels, consumption content, consumption patterns, and consumption concepts, known as the Consumption Upgrading (CSU) evaluation index system; it comprises 26 indicators in five areas, as shown in Table 2. The system aims to comprehensively reflect the development of consumption object content, consumption levels, consumption concepts, and patterns from low to high levels.

**Table 2.** Comprehensive indicator system of consumption upgrading.

| Guideline Level | Specific Indicators | Indicator Description/Measurement | Indicator Attributes | Indicator Weights |
|---|---|---|---|---|
| Total social consumption | Consumption rate | Consumption level/GDP | + | 0.019 |
| | Total social consumption | Total retail sales of social consumer goods | + | 0.065 |
| | Growth rate of total social consumption | Total retail sales of consumer goods growth rate | + | 0.003 |
| | Number of workers in the tertiary sector | - | + | 0.018 |
| Consumption level of residents | Per capita consumption expenditure | Consumer spending/regional year-end population | + | 0.036 |
| | Consumption growth rate | Growth rate of consumer spending per capita | + | 0.007 |
| | Urban to rural consumption ratio | Ratio of per capita consumption expenditure of urban and rural residents | − | 0.012 |
| Consumer content | Per capita consumption expenditure on household equipment and services | - | + | 0.031 |
| | Per capita consumption expenditure on transport and communications | - | + | 0.043 |
| | Health care consumption expenditure per capita | - | + | 0.027 |
| | Per capita consumption expenditure on education, culture and entertainment | - | + | 0.033 |
| | Other consumption expenditure per capita | - | + | 0.037 |
| | Developmental consumption as a percentage | (Consumption expenditure on household equipment and services + expenditure on transport and communication)/total consumption expenditure | + | 0.018 |
| | Percentage of consumption for enjoyment | (Household consumption expenditure on health care + education, culture and recreation + other consumption expenditure)/total consumption expenditure | + | 0.008 |

**Table 2.** *Cont.*

| Guideline Level | Specific Indicators | Indicator Description/Measurement | Indicator Attributes | Indicator Weights |
|---|---|---|---|---|
| Consumer content | Consumption structure upgrading | Share of subsistence consumption + share of developmental consumption × 2 + share of enjoyment consumption × 3 | + | 0.007 |
| | Engel coefficient | Food, tobacco and alcohol expenditure/consumption expenditure | − | 0.011 |
| | Car ownership | Private car ownership/year-end population | + | 0.038 |
| Consumption pattern | Total postal and telecommunications services | - | + | 0.008 |
| | Total express delivery per capita | Total number of couriers/area year-end population | + | 0.194 |
| | Telephone penetration rate | Telephone penetration rate | + | 0.024 |
| | Total restaurant and accommodation business | Turnover of catering businesses + turnover of accommodation businesses | + | 0.088 |
| | Service Levels in Catering and Accommodation | (Number of legal entities in the accommodation sector + number of legal entities in the catering sector)/year-end population | + | 0.08 |
| | Number of travel agents | - | + | 0.092 |
| Consumer Philosophy | Low carbon consumption | City gas penetration rate | + | 0.003 |
| | Green Travel | Number of public transport rides per capita | + | 0.045 |
| | Risk management | Premium density | + | 0.052 |

### 3.1.3. Evaluation Methods and Data Sources

To ensure objectivity in the determination of indicator weights and address the issues of crossover and superposition of information among composite indicators in the multi-level evaluation indicator system of residents' consumption capacity and consumption upgrading, this study employed the min-max method of efficacy coefficients to standardize the indicators. Subsequently, the entropy method was used to assign weights to each specific indicator, and the residents' consumption capacity and consumption upgrading were evaluated using the linear weighted summation method. This approach has been widely used in various countries and disciplines, including the evaluation of basic public service provision [28], consumption upgrading [8], and green industry development [29].

This empirical investigation employs provincial-level sample data comprising a total of 403 research samples from 31 provinces, municipalities, and autonomous regions (here-inafter referred to as "provinces") in mainland China spanning from the year 2008 to 2020. The data utilized in this study were primarily obtained from reputable sources such as the China Statistical Yearbook, China Education Statistical Yearbook, China Environmental Statistical Yearbook, China Health and Family Planning Statistical Yearbook, statistical yearbooks of the 31 provinces, and the National Bureau of Statistics website. Years of education per capita and industrial structure indicators were computed from the original data, whereas missing data were supplemented using linear interpolation or extrapolation.

### 3.2. Spatial Distribution and Regional Variation Decomposition Methods

To explore the spatial distribution and inter-regional differences in consumption capacity and consumption upgrading, this study uses Moran's I index and the Dagum Gini coefficient to carry out the analysis. Among them, Moran's I index can effectively measure the similarity of observations in spatially neighboring regions, which helps to observe the

spatial distribution characteristics and interactive development patterns of indicators. It is calculated as follows:

$$Moran\prime I = \frac{\sum\limits_{i=1}^{n}\sum\limits_{j=1}^{n}\omega_{ij}(y_i - \overline{y})(y_j - \overline{y})}{S^2\sum\limits_{i=1}^{n}\sum\limits_{j=1}^{n}\omega_{ij}} \tag{1}$$

In Equation (1), $S^2 = \frac{1}{n}\sum\limits_{i=1}^{n}(y_i - \overline{y})^2$; $\overline{y} = \frac{1}{n}\sum\limits_{i=1}^{n}y_i$; $y_i(y_j)$ is the CSP (CSU) evaluation indices of the $i(j)$ province; $\omega_{ij}$ is the $(i,j)$ element of the gravitational space weight matrix $W$, when $j \neq i$, $\omega_{ij} = \frac{GDP_i^* GDP_j^*}{d_{ij}^2}$—otherwise $\omega_{ij} = 0$—where $d_{ij}$ is the straight-line distance between the capital city of province $i$ and the capital city of province $j$, and $GDP^*$ is the economic strength of each province measured by the average GDP during the sample period.

The Dagum Gini coefficient and its decomposition method have been applied to the analysis of regional differences in consumption capacity and consumption upgrading. The Dagum Gini coefficient can effectively solve the problem of overlapping data and sources of variation in research samples, compensating for the shortcomings of traditional methods such as the Gini coefficient and the Theil index, and is now widely used to explore the relative differences between regions in different areas. Dagum [30] decomposes the overall Gini coefficient $G$ by sample into three components: intra-regional variation $G_w$, net inter-regional variation $G_{nb}$, and hyper-variance density $G_l$, which satisfy $G = G_w + G_{nb} + G_l$, as defined in Equation (2).

$$G = \sum_{j=1}^{k}\sum_{h=1}^{k}\sum_{i=1}^{n_j}\sum_{r=1}^{n_j}|y_{ji} - y_{hr}|/(2n^2\overline{y}) \tag{2}$$

In Equation (2), $y_{ji}(y_{hr})$ is the composite CSP (CSU) index of $i(r)$ provinces within the $j(h)$ region, $\overline{y}$ is the mean CSP (CSU) index for each province, $n$ represents the number of provinces in the sample, $k$ is the number of regions divided, $n_j(n_h)$ is the number of provinces within the $j(h)$ regions, and the Dagum Gini coefficient is calculated in detail in the relevant literature [30].

*3.3. Spatial Interaction Analysis Model*

As inter-regional interactions become increasingly close, both CSP and CSU in one province may have spatial spillover to neighboring provinces through direct or indirect channels, such as imitation and demonstration, leading to inter-provincial interaction effects. Although traditional VAR models can portray the interaction effects between variables, it is difficult to characterize the spatial dynamics. Therefore, this study used the SPVAR model to test the spatial dynamic interaction effect of CSP and CSU, with the model form shown in Equation (3).

$$Y_{it} = \beta Y_{it-1} + \lambda W Y_{it-1} + \mu_i + \eta_t + \varepsilon_{it} \tag{3}$$

$$Y_{it} = \begin{bmatrix} CSP_{it} \\ CSU_{it} \end{bmatrix}, \beta = \begin{bmatrix} \beta_{11} & \beta_{12} \\ \beta_{21} & \beta_{22} \end{bmatrix}, \lambda = \begin{bmatrix} \lambda_{11} & \lambda_{12} \\ \lambda_{21} & \lambda_{22} \end{bmatrix}, \mu_i = \begin{bmatrix} \mu_{1i} \\ \mu_{2i} \end{bmatrix}, \eta_t = \begin{bmatrix} \eta_{1t} \\ \eta_{2t} \end{bmatrix}$$

In Equation (3), the subscripts $i(i = 1, \cdots, 31)$ and $t(t = 1, \cdots, 13)$ denote province $i$ and year $t$, respectively; $Y_{it}$ is the vector form of the endogenous variables CSP and CSU; $Y_{it-1}$ is the vector form of the endogenous variables lagging one period; $W$ is the matrix of gravitational space weights; $\beta$ is the matrix of coefficients of the endogenous variables lagging one period; $\lambda$ is the matrix of coefficients of the endogenous variables lagging one period in time; $\mu_i$ and $\eta_t$ denote the time-invariant province fixed effects and the time-invariant individual fixed effects, respectively, and $\varepsilon_{it}$ is the spatial error term.

## 4. Evaluation of Residents' Consumption Capacity and Consumption Upgrading and Its Spatial Differentiation

*4.1. Description of Research*

To examine the spatial and temporal changes in residents' consumption capacity and consumption upgrading, we calculated the average values of CSP and CSU for each cycle within the framework of China's economic planning cycle. Subsequently, we analyzed the evolution of consumption capacity and consumption upgrading from both the horizontal and vertical perspectives.

In the middle and late stages of the Eleventh Five-Year Plan (2008–2010), the CSP in the eastern region (2.429) was higher than that of the entire country (1.763) and the central (1.603) and western (1.259) regions. The average value of the CSP in the central and western regions was lower than that of the entire country, and the average value in the central region was higher than that of the western region. This distribution is closely related to internal and external factors such as factor endowments and economic development levels in each region. During the 12th Five-Year Plan period, CSP experienced a rapid increase of over 36% compared with the middle and late stages of the 11th Five-Year Plan. However, the growth trend differed between regions, with the growth rates of the western and central regions reaching 44.482% and 37.766%, respectively, which were significantly higher than the average growth rate of 29.388% in the eastern region. Nevertheless, the decreasing growth rates in the west, center, and east did not significantly impact the spatial pattern of CSP. In the 13th Five-Year Plan period, the CSP levels of all provinces in China increased further while maintaining the previous spatial pattern, and the growth rates in the west and central regions continued to exceed those in the east. Generally speaking, the consumption capacity of residents in each province is on a rising trend, showing an orderly pattern of decrease in the east, center, and west. However, the regional growth rate shows the opposite trend, with the central and western regions "catching up" with the east. The consumption capacity of residents in each province may gradually converge to the same level in the future.

As for the CSU, the range of variation in the CSU in the mid to late 11th Five-Year Plan is [0.508, 3.841], with a mean value of 1.300 and a standard deviation of 0.674, indicating that consumption upgrading in China was still at a low level at that stage and that there were significant differences between provinces and regions. Among them, eight provinces exceed the national average of CSU, all of which are located in the eastern region, with Beijing, Shanghai, and Guangdong ranking in the top three of CSU (7.566, 5.965, and 4.445 times higher, respectively, than Tibet at the bottom). During the 12th Five-Year Plan period, the CSU continued to maintain the spatial pattern of the east being higher than the central and west, but the opposite growth trends in the east, central, and west helped narrow the differences between regions, with the growth rates in the west, central, and east being 54.718%, 49.908%, and 40.425%, respectively. Thus, the ratio of CSU values in Beijing, Shanghai, Guangdong, and Tibet fell to 6.270, 4.940, and 4.126, respectively. During the 13th Five-Year Plan period, the spatial distribution pattern of CSU changed less and continued to maintain the trend of regional differentiated growth, which means that the growth rate of CSU continued to maintain the trend of decrease in the west, middle, and east. The provinces with lower CSU gradually caught up with the high-development provinces because they have higher growth rates, and the gap between provinces gradually narrowed.

Moreover, a comparative analysis of consumption capacity and consumption upgrading dynamics indicated that CSP and CSU exhibited a consistent upward trend during the observation period. Further examination of the lagged structure of consumption capability and consumption upgrading reveals that CSU generally lags behind CSP, but the gap between the two is diminishing. This implies that a stable positive feedback relationship integrating consumption capability and consumption upgrading may not have emerged in China and that it remains necessary to promote consumption upgrading through residents' consumption capability. However, since the 12th Five-Year Plan period, the trend of the CSU lagging behind the CSP in Beijing has been shifting, with the CSU gradually taking

the lead. This trend extended to the Beijing, Shanghai, and Zhejiang provinces during the 13th Five-Year Plan. These observations suggest that consumption upgrading is not solely driven by consumption capacity but may also benefit from endogenous drivers of consumption upgrading. Furthermore, the pulling effect of consumption upgrading on residents' consumption capacity is increasing. This shift is primarily due to changes in residents' consumption expectations, patterns, and concepts, with a greater focus on the pursuit of consumption quality after their "quantity" needs have been met. This has accelerated the expansion and upgrading of consumption, with the CSU contributing to the high growth of the economy and a new choice for the industrial chain division of the labor system. This is an important factor in the growth of residents' consumption capacity.

*4.2. Spatial Agglomeration Patterns of CSP and CSU*

　　The results presented in Table 3 reveal that Moran's I indices of CSP and CSU in all years are positive and significant at the 1% level, indicating that CSP and CSU in China exhibited non-random spatial distribution and significant spatial clustering. Specifically, the spatial clustering of CSP and CSU in China suggests a "clustering of things" phenomenon, whereby the consumption capacity and upgrading of residents in each province are closely related to those of neighboring regions. This finding highlights the importance of considering the spatial dependence of consumption patterns when analyzing consumer behavior in China and shows that Moran's I index of CSU has been on a declining trend during the observation period, indicating that the spatial correlation between consumption upgrading in China's provinces has been decreasing in recent years. Despite some fluctuations, the overall trend has relatively rapid growth, suggesting that the spatial spillover effect of residents' consumption capacity is increasing.

**Table 3.** Moran's I index for CSP and CSU.

| Year | CSP | | CSU | |
| --- | --- | --- | --- | --- |
| | I | Z | I | Z |
| 2020 | 0.236 *** | 3.007 | 0.253 *** | 3.246 |
| 2019 | 0.213 *** | 2.752 | 0.251 *** | 3.242 |
| 2018 | 0.233 *** | 2.966 | 0.230 *** | 3.03 |
| 2017 | 0.262 *** | 3.28 | 0.236 *** | 3.128 |
| 2016 | 0.259 *** | 3.252 | 0.228 *** | 3.058 |
| 2015 | 0.285 *** | 3.532 | 0.205 *** | 2.826 |
| 2014 | 0.288 *** | 3.581 | 0.196 *** | 2.733 |
| 2013 | 0.315 *** | 3.92 | 0.228 *** | 3.121 |
| 2012 | 0.318 *** | 3.907 | 0.215 *** | 2.982 |
| 2011 | 0.346 *** | 4.276 | 0.232 *** | 3.209 |
| 2010 | 0.354 *** | 4.368 | 0.215 *** | 3.047 |
| 2009 | 0.380 *** | 4.715 | 0.192 *** | 2.809 |
| 2008 | 0.342 *** | 4.351 | 0.185 *** | 2.706 |

Note: *** denote significant at the 1% level.

　　Further, the local spatial clustering characteristics revealed by the Moran's scatter plot indicate that the local Moran's I index scatter is predominantly concentrated in the first and third quadrants of the CSP and CSU, implying the prevalence of "high–high" or "low–low" clustering. Only a few provinces exhibit negative outlier "high–low" or "low–high" clustering, and the clustering characteristics remain stable over the observation period. The "high–high" clusters are relatively scarce and are mainly located in the eastern coastal or central fast-rising regions, where economic growth is rapid, the commodity economy is thriving, residents have higher income levels, infrastructure is better, and

social security and welfare are more comprehensive. Moreover, residents in these regions have a higher propensity to consume, driven by higher education levels, optimistic future income expectations, a greater focus on experiential consumption, and a willingness to try new products. This has led to increased spending power and accelerated consumption upgrading, potentially having a "demonstration effect" on neighboring provinces. The "low–low" clustering provinces are predominantly located in the central and western regions, where the economic base is weak, residents' income is relatively low, and social security and infrastructure require further improvement. Additionally, related consumer support facilities, such as the Internet and logistics, are underdeveloped, and the level of consumer spending is low, with a relatively homogeneous consumption structure and traditional consumption concepts. This area has gradually become a "depression" for CSP and CSU. The "high–low" ("low–high") provinces are primarily Shanxi, Inner Mongolia, Jiangxi, Guangdong, and Hainan, either located around "high–high" agglomerations, where both CSP and CSU have more potential for improvement, or where the spatial radiation effect of residents' consumption capacity and consumption upgrading is smaller. The improvement of residents' consumption capacity and consumption upgrading in the province does not have a driving effect on neighboring provinces and cities, but rather there is a certain degree of polarization. The above analysis suggests that there is similar spatial heterogeneity in the distributions of CSP and CSU in China, indicating a significant positive interaction between the two. Moreover, the agglomeration pattern is highly similar to the traditional division of the three major zones in China, necessitating the division of the sample provinces into three major regions, east, central, and west, for further regional analysis.

### 4.3. Regional Disparities in CSP and CSU

To investigate the magnitude and determinants of regional differences in CSP and CSU in China, this study employs the Dagum Gini coefficient and decomposition approach for scientific measurement and decomposition. The measurement results are shown in Figures 1 and 2.

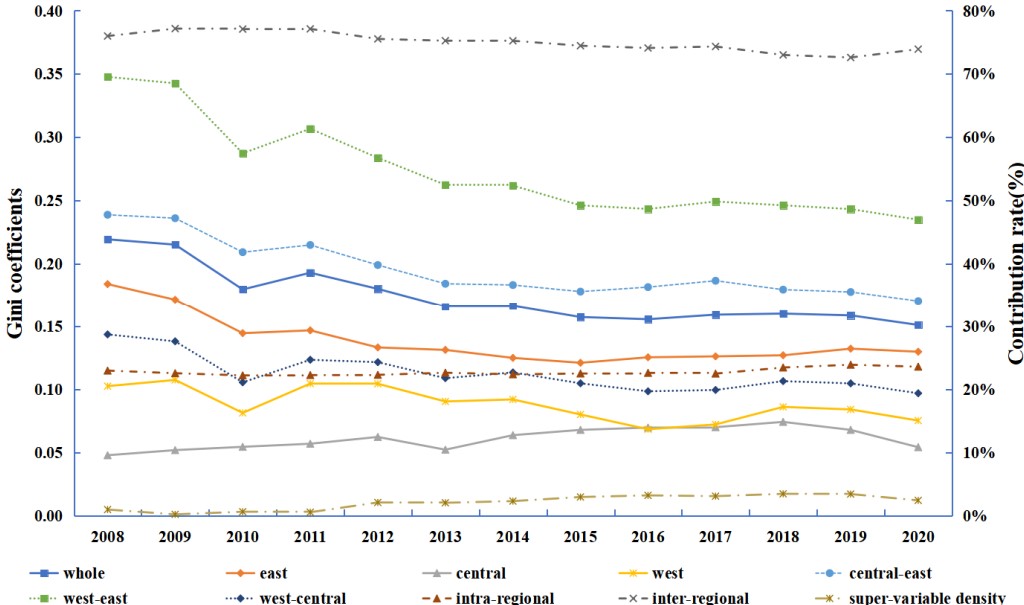

**Figure 1.** The Dagum Gini coefficient and decomposition results of CSP.

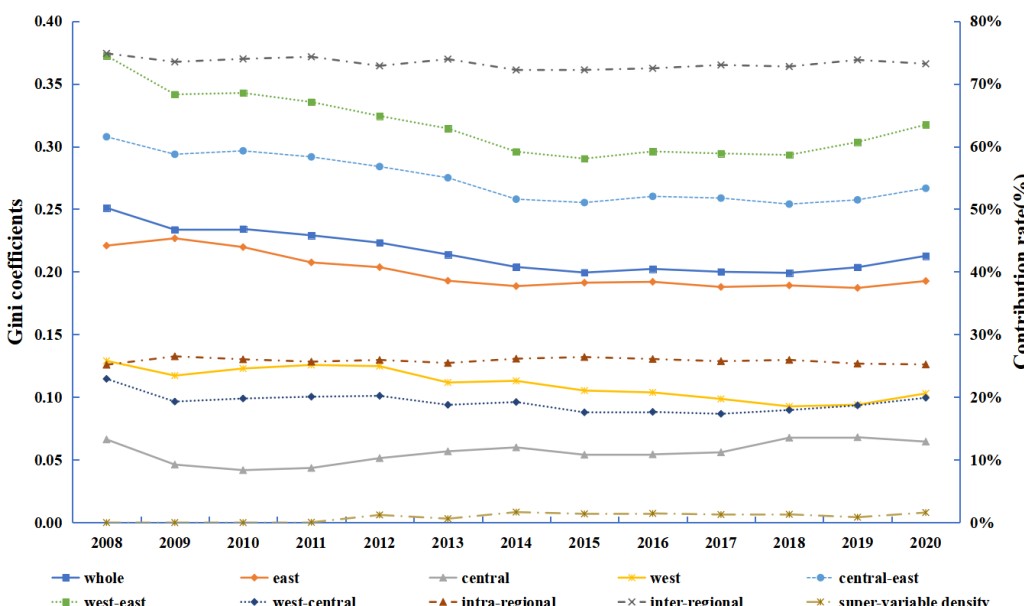

**Figure 2.** The Dagum Gini coefficient and decomposition results of CSU.

Figures 1 and 2 illustrate the temporal evolution of the spatial disparities between CSP and CSU across China during the observation period. The overall Gini coefficient reveals that the spatial gaps between CSP and CSU exhibit a similar trend, displaying a gradual decline of 31.14% and 15.253%, respectively. This suggests that the disparity between CSP and CSU is gradually diminishing in all regions of China. Specifically, CSP and CSU showed persistent decline from 0.220 and 0.251 to 0.166 and 0.214 during the period 2008–2013, respectively, followed by a relatively stable period from 2014 to 2018. After 2018, the overall difference between CSP and CSU diverged, with CSP continuing to decline and reaching a new low of 0.151 in 2020. However, the CSU exhibited an upward trend, with the overall difference increasing from 0.199 in 2018 to 0.213, reflecting a 6.802% increase. Moreover, the Gini coefficient of CSU was higher than that of CSP in all years, indicating that regional disparities in consumption upgrading were greater than those in spending power. This is attributed to the combined effect of steady economic development and policies that benefit people. These policies have significantly reduced interregional disparities in income, employment, healthcare, cultural quality, and other livelihood components, thereby enhancing the ability and willingness of residents to consume. However, consumption upgrading primarily depends on changes in consumption patterns, tendencies, and concepts, which tend to exhibit inertia and take a longer time to converge than the rapid reduction of inter-regional gaps in consumption capacity.

The mean values of the Gini coefficient of the CSP, which reflects the level of intra-regional variation, were 0.138, 0.061, and 0.089 for the eastern, central, and western regions, respectively, during the observation period. These values suggest that the eastern region has the largest intra-regional variation, followed by the western region, whereas the central region has the smallest intra-regional variation. The Gini coefficient in the eastern region exhibited a persistent decline, whereas that in the western region experienced a slight decline with frequent fluctuations. The central region displayed an upward and then a downward trend, and an overall increasing trend. The Gini coefficient increased significantly in the central region, decreased more in the eastern region, and changed less in the western region. Furthermore, based on the trend of the Gini coefficient between CSP regions in China during the sample period, the mean value of the Gini coefficient between east–west region was 0.274, indicating the highest level of variation between the regions. The east–central region had the second-highest level of variation, while the central–west region had the least variation. The trend of change indicated an evolutionary trend of spatial divergence between the three major regions, with the largest decrease in the inter-regional Gini coefficient in

the central–western region, demonstrating a decrease of 32.421%. This is followed by the east–west inter-regional divergence, which also fell by 32.412%, whereas the east–central inter-regional divergence Gini coefficient decreased by 28.540%. These results suggest that spatial heterogeneity still exists in the CSP in China, while the differences between the three major regions and within the eastern and western regions demonstrate a gradual decrease, indicating a convergent development pattern. However, an increase in the degree of divergence may have occurred within the central region.

The intra-regional Gini coefficient of the CSU shows a higher degree of intra-regional variation in the east and west regions (the mean Gini coefficients for the east and west regions are 0.200 and 0.111, respectively) and a lower degree of non-parity within the central region (the mean Gini coefficient is 0.056). The evolution of regional Gini coefficients revealed that both the east and west regions experienced a "downward-crossing-slightly upward" trend, with the Gini coefficients decreasing by 12.773% and 20.156%, respectively, during the observation period. In contrast, the Gini coefficient of the central region showed a fluctuating horizontal extension. According to the characteristics of the inter-regional Gini coefficient of the CSU depicted in Figure 1, the mean value of the Gini coefficient between the east and west regions was 0.317 during the observation period, with the greatest difference between the two regions, with the Gini coefficient first showing a fluctuating downward trend, and then a "tail", but with an overall decrease of 14.767%; followed by the east–central region, where the average value of the Gini coefficient is 0.274, and the trend is basically in line with the east–west region, but the magnitude of the change is smaller, with a drop of 13.351% during the observation period. The least regional variation is in the central–western region, with Gini coefficient values ranging from 0.087 to 0.115, showing a "falling–rising–falling–rising–falling" trend with frequent horizontal fluctuations. On the whole, the CSU also has obvious spatial divergences, and the three major regions and regions within the east and west have convergent development characteristics, while the degree of divergence in the central region is basically at a relatively stable level.

From the decomposition values of the Gini coefficients of the CSP and CSU and their contributions, the most important cause of inter-regional differences in the CSP and CSU is inter-regional differences. The contribution of inter-regional differences to the overall disparity in both CSP and CSU remained relatively stable from 2008 to 2020, with persistent horizontal trends and average contribution rates of 75.063% and 73.300%, respectively. This is much higher than the combined contribution of intra-regional differences and super-variable density, which means that for the overall regional differences in CSP and CSU to be consistently and steadily mitigated, there is an urgent need to formulate and implement regional synergistic development policies to effectively narrow the gap between regional residents' consumption capacity and consumption upgrading, in order to effectively promote consumption expansion and upgrading on a sustainable basis. For the CSP, the within-group gap contributed between 22% and 24% of the overall variation, with a weak and steady upward trend. In contrast, the within-group gap contributed between 25% and 27% of the overall variation in CSU, with a weak "M-shaped" change, suggesting that the evolution of CSP and CSU within a region alone is generally stable. The contribution of hypervariable density, which measures the extent of cross-overlap in inter-regional variation, to the overall variation in CSP and CSU during the observation period was relatively small, in the ranges [0.246%, 3.498%] and [0%, 1.670%], respectively. However, both tended to increase, indicating that the inter-regional cross-overlap phenomenon gradually emerged and the gap between the CSP and CSU regions weakened. It can be seen that the spatial divergence of CSP and CSU in China is importantly due to inter-regional differences, followed by intra-regional differences, while the contribution of hypervariable density to the overall differences is relatively small.

## 5. Study on the Spatial Dynamic Relationship between Consumption Capacity and Consumption Upgrading

Based on the SPVAR model and estimation method set out above, before testing the interaction between consumption capacity and consumption upgrading, and the inter-provincial spillover effect, the smoothness and spatial correlation of the series of variables in the model need to be tested.

### 5.1. CSP and CSU Smoothness Tests

To mitigate issues such as "pseudo-regression" and ensure the reliability of the SPVAR model when examining the spatial dynamic interaction between CSP and CSU, it is imperative to assess the smoothness of the two series. We conducted panel data LLC, Breitung IPS, ADF, and PP tests on both the CSP and CSU series, considering robustness as a key criterion. The results indicate that the original CSP series exhibits significant smoothness based on the intercept-containing test, at least at the 10% level. Similarly, the original CSU series is also a smooth series with both intercepts and trends passing the 10% significance level. This finding suggests a stable long-term equilibrium relationship between CSP and CSU.

### 5.2. Panel Granger Causality Test for CSP and CSU

In order to test whether there is a statistically significant bootstrap relationship between the CSP and CSU series, this paper will conduct a panel Granger test on both variables for confirmation. Given that both the CSP and CSU original series are smooth series, this paper refers to Hsiao et al. [31] and performs a Wald test to determine Granger causality between the two variables based on estimating a panel VAR fixed effects model and a random effects model.

To examine whether a statistically significant bootstrap relationship exists between the CSP and CSU series, we conducted a panel Granger test on the two variables. Because both the CSP and CSU original series are smooth, following Hsiao et al. [31], we performed a Wald test to determine the Granger causality between the two variables by estimating a panel VAR fixed effects model and a random effects model. The optimal lag order of the PVAR model, as determined by the Akaike information criterion (AIC) and Bayesian information criterion (BIC), should be lagged to order 1. Table 4 presents the results of the panel Granger test. The binary fixed effects model (1) shows that the original hypothesis that CSU is not the Granger cause of CSP is rejected at the 1% level, suggesting that consumption upgrading is an important cause of the increase in consumption capacity. The binary fixed effects model (2) shows that the alternative hypothesis that CSP is the Granger cause of CSU is accepted at the 5% significance level. This indicates that an increase in residents' consumption capacity has a facilitating effect on consumption upgrading in the medium-to-long term.

**Table 4.** Panel Granger causality test results.

| Serial Number | Panel Type | Explained Variables | Constants (C1) | CSP(-1) (C2) | CSU(-1) (C3) | Hausman Testing | Wald Test | |
|---|---|---|---|---|---|---|---|---|
| | | | | | | | H0 | F-Statistic |
| (1) | Binary fixed Effect | CSP | 0.033 (0.008) | 0.812 (0.035) | 0.100 (0.027) | 35.45 *** | C3 = 0 | 13.19 *** |
| | | | | | | | CSU is the Granger reason for CSP | |
| (2) | Binary fixed Effect | CSU | −0.050 (0.008) | 0.179 (0.035) | 0.928 (0.027) | 6.91 ** | C2 = 0 | 26.22 *** |
| | | | | | | | CSP is the Granger reason for CSU | |

Note: ***, ** denote significant at the 1% and 5% levels, respectively, with the standard deviation of the corresponding estimates in parentheses.

### 5.3. Estimation of SPVAR Models for CSP and CSU

It is worth noting that SPVAR, as an extension of the VAR model, does not have significant practical implications for the economic interpretation of individual parameter estimates, making it challenging to evaluate the model by estimating coefficients. However,

the corresponding impulse function can be used to explore the relationships between the variables [32]. Therefore, the results of the model parameter estimation are only presented (Table 5), this study calculates the spatiotemporal impulse response function of the model to simulate the interaction effect between CSP and CSU within each province as well as the spatial spillover effect across provinces and regions.

**Table 5.** SPVAR model estimation results.

| Variables | CSP | CSU |
|---|---|---|
| CSP(−1) | −0.203 | 0.154 |
| | (0.114) | (0.120) |
| CSU(−1) | 0.171 | 0.192 |
| | (0.103) | (0.109) |
| CSP(−1) spatial lag term | 0.092 | 0.437 |
| | (0.146) | (0.14) |
| CSU(−1) spatial lag term | 0.004 | 0.272 |
| | (0.132) | (0.127) |
| Adj-R2 | 0.904 | 0.927 |

Note: Standard deviations of the corresponding estimates are in parentheses.

### 5.4. Spatio-Temporal Impulse Response Analysis of CSP and CSU

Compared with the panel VAR model, the impulse response of the SPVAR model is reflected in both the temporal and spatial dimensions. In addition to the two sources of shocks brought about by the CSP and CSU, disturbances occurring in different cross-sections will also lead to distinct response effects. This study investigates 31 cross-sections that provide 62 different sources of shocks. Owing to space constraints, this study selected Beijing, one of the first international consumer center cities to be cultivated and built in China, as a representative example to examine the spatial dynamic interaction between CSP and CSU in detail.

#### 5.4.1. Internal Shock Transmission Mechanisms of CSP and CSU

Figure 3a,b show the dynamic response process of Beijing's own two variables after adding a unit standard deviation positive shock to CSP and CSU, respectively. When a positive unit standard deviation was applied to the CSP of Beijing, the CSP showed an initial positive response in period 1, followed by a negative response in period 2. Subsequently, CSP experiences a small increase and again becomes positive, after which it converges to zero during the fluctuation starting in period 4. However, when a positive unit standard deviation shock is applied to the CSU, the shock effect initially leads to an increase in the CSU. However, the effect gradually diminishes and eventually converges to zero during the fluctuation. Additionally, the CSP response to a positive unit standard deviation shock from the CSU initially rises and peaks in period 2 before beginning to fall, and gradually decays to zero from period 7.

In general, CSP and CSU in Beijing exhibit inherent mechanisms that drive their operations. Additionally, there exists a positive interaction effect between CSP and CSU that promotes mutual growth, referred to as the "positive feedback effect." However, the translation of potential consumption expansion and upgrading demand into real demand occurs rapidly, in contrast to the thread of consumption upgrading, which stems primarily from economic and social development. This leads to a longer time lag in increasing the population's consumption capacity, which is likely to promote consumption upgrading. Although consumption upgrading has a positive effect on consumption capacity, its impact is relatively weak and exhibits a certain time lag.

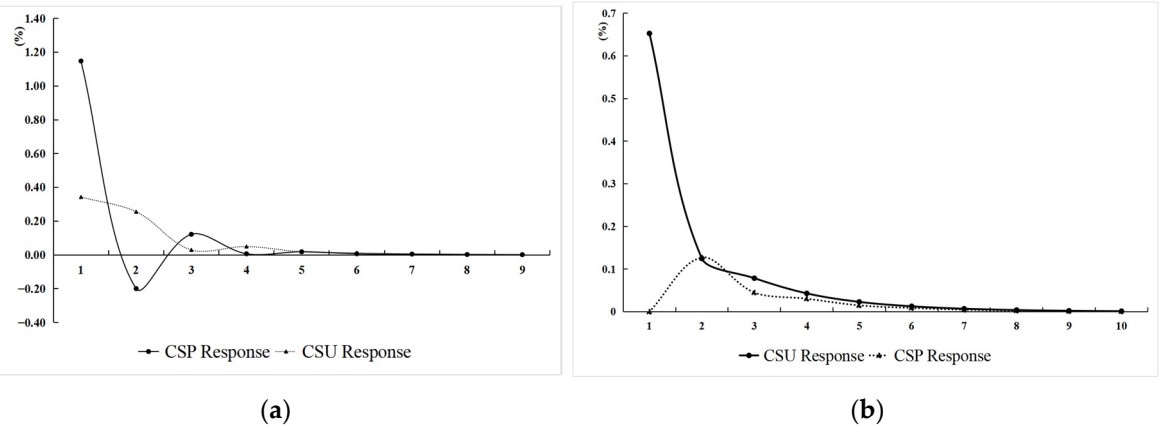

<div align="center">(<b>a</b>)                   (<b>b</b>)</div>

**Figure 3.** Interaction between CSP and CSU in Beijing. (**a**) Pulse source CSP, (**b**) Pulse source CSU.

5.4.2. Spatial Spillover Effects of CSP and CSU

Furthermore, this study analyzes the cross-provincial spillover effects of CSP and CSU from the dynamic response results, starting from the impact of Beijing's CSP and CSU unit standard deviation shocks on other provinces. According to the impulse response results, the responses of CSP and CSU to Beijing CSP and CSU shocks are similar across provinces, with the impulse responses generally showing a "Rising–falling–stabilising" pattern, but there is significant heterogeneity in the degree of impulse responses across provinces.

The cumulative effects of positive shocks from the Beijing CSP and CSU unit standard deviations stabilized (10 periods) are shown in Figures 4 and 5 for each province. Overall, the cumulative effect of positive shocks to CSP or CSU unit standard deviation from Beijing on CSP and CSU is positive in all provinces, and the cumulative impact on CSP in other provinces is generally higher than that on CSU, suggesting that increased consumption capacity or consumption upgrading in Beijing is more likely to drive the growth of consumption capacity in neighboring provinces. The five most significantly affected provinces and cities are Tianjin, Hebei, Inner Mongolia, Shandong, and Shanxi. The continuous deepening of the Beijing–Tianjin–Hebei cooperative development has accelerated the integration of market sharing, consumption imitation, cross-regional consumption, and policy incentives between them. The economic interaction between the regions has become closer, with consumption capacity and consumption upgrading between the regions influencing and interacting with each other, thus giving Beijing a strong spatial spillover to Tianjin and Hebei. Although Inner Mongolia, Shandong, and Shanxi are geographically distant from Beijing, the increasing improvement of inter-regional transport and communication infrastructure has led to closer spatial links between people, material, and information flows across the region. The integration of income, consumption interactions, and consumption attitudes and patterns between Beijing and the neighboring provinces has continued to penetrate, thus allowing Beijing to exert a greater influence on these three provinces as well.

The provinces of Zhejiang, Shanghai, Anhui, Guangxi, and Hainan experienced the least impact from Beijing's shocks. Geographically, Guangxi and Hainan are located in southern China far from Beijing, which limits their economic and consumption interactions. However, their proximity to developed regions such as Guangdong, Macau, and Hong Kong has a substitution effect on the spatial interactions between Guangxi, Hainan, and Beijing to a certain extent. Shanghai, Zhejiang, and Anhui are located in the Yangtze River Delta region, a key growth pole for China's economy and consumption. The region has international or regional consumption centers, such as Shanghai, Hangzhou, and Hefei, which exercise strong control over goods and services in circulation, forming a specific and stable consumption spatial interaction mechanism in the region. This mechanism weakens or even counteracts the impact of Beijing, and the relatively large spatial distance limits the effectiveness of Beijing's policies.

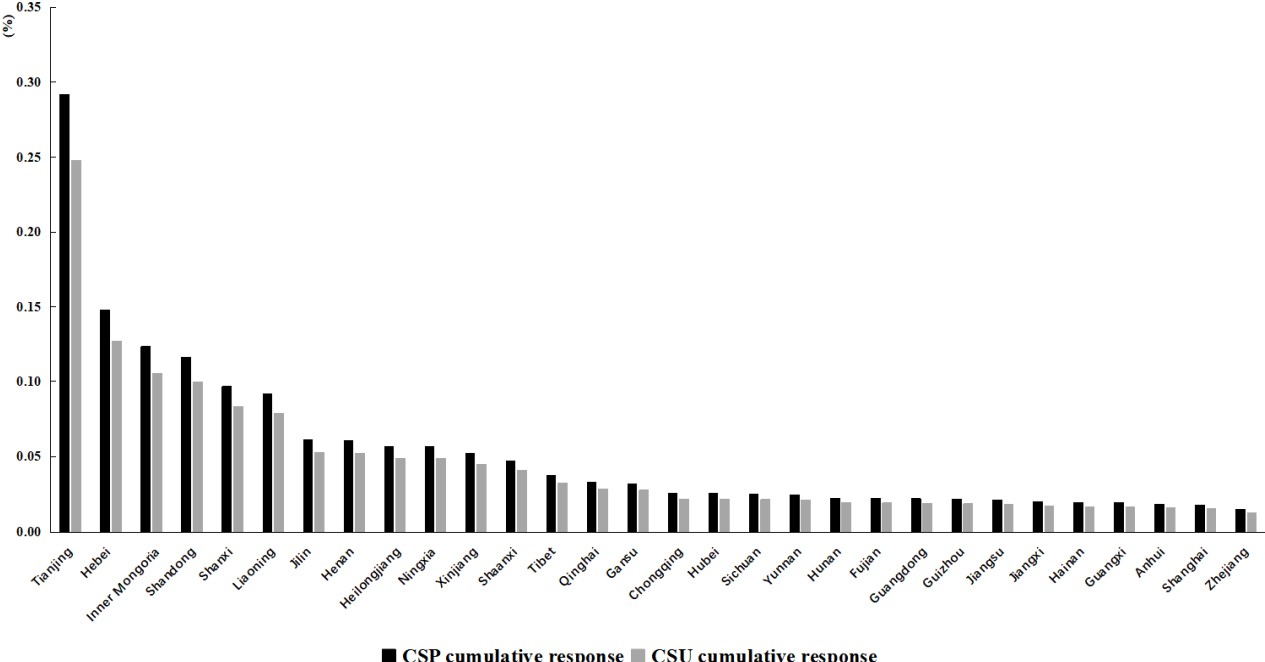

**Figure 4.** Cumulative impact of CSP shocks in Beijing on other provinces.

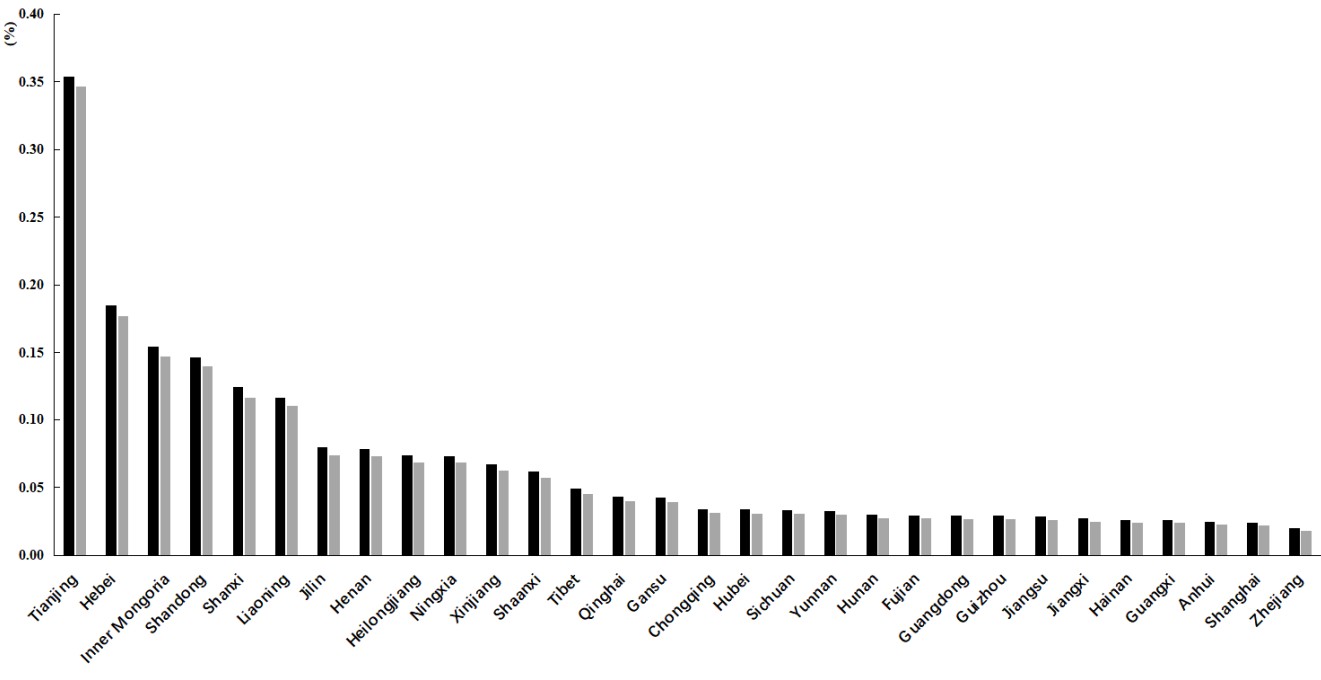

**Figure 5.** Cumulative impact of CSU shocks in Beijing on other provinces.

Generally, the study findings reveal a clear "center–periphery" spillover effect of consumption capacity and consumption upgrading shocks in China. The shock effect is stronger when geographical proximity or economic ties are closer, and weaker when they are further apart. Moreover, higher levels of one's own consumption capacity and consumption upgrading result in a stronger ability to resist shocks and lower susceptibility to their impact. Conversely, lower levels of consumption capacity and consumption upgrading result in greater susceptibility to shocks.

### 5.4.3. The Reverse Spatial Spillover Effects of CSP and CSU

The preceding analysis shows that the CSP and CSU shocks in Beijing affected the evolution of CSP and CSU in other provinces. Conversely, changes in CSP and CSU in other provinces can trigger shock effects on Beijing. To facilitate comparison, Figures 6 and 7 show the cumulative impact of positive unit standard deviation changes in CSP and CSU in other provinces on Beijing.

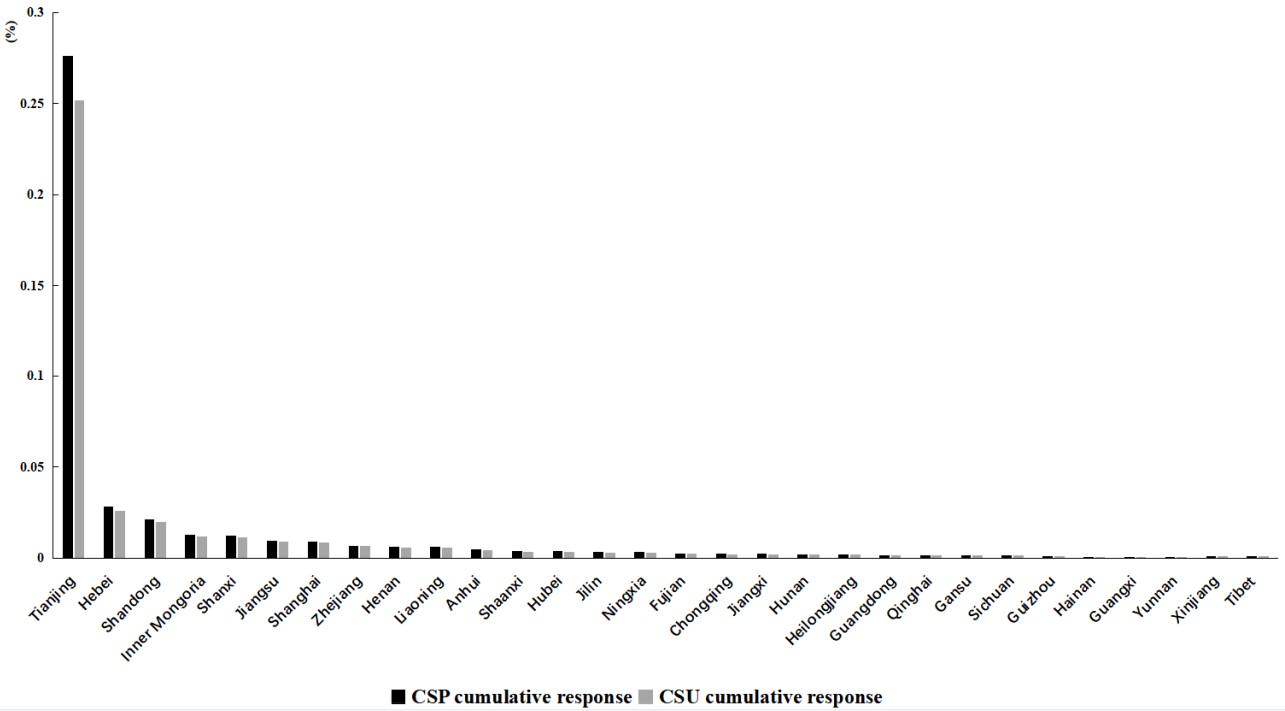

**Figure 6.** Cumulative impact of CSP shocks in other provinces on Beijing.

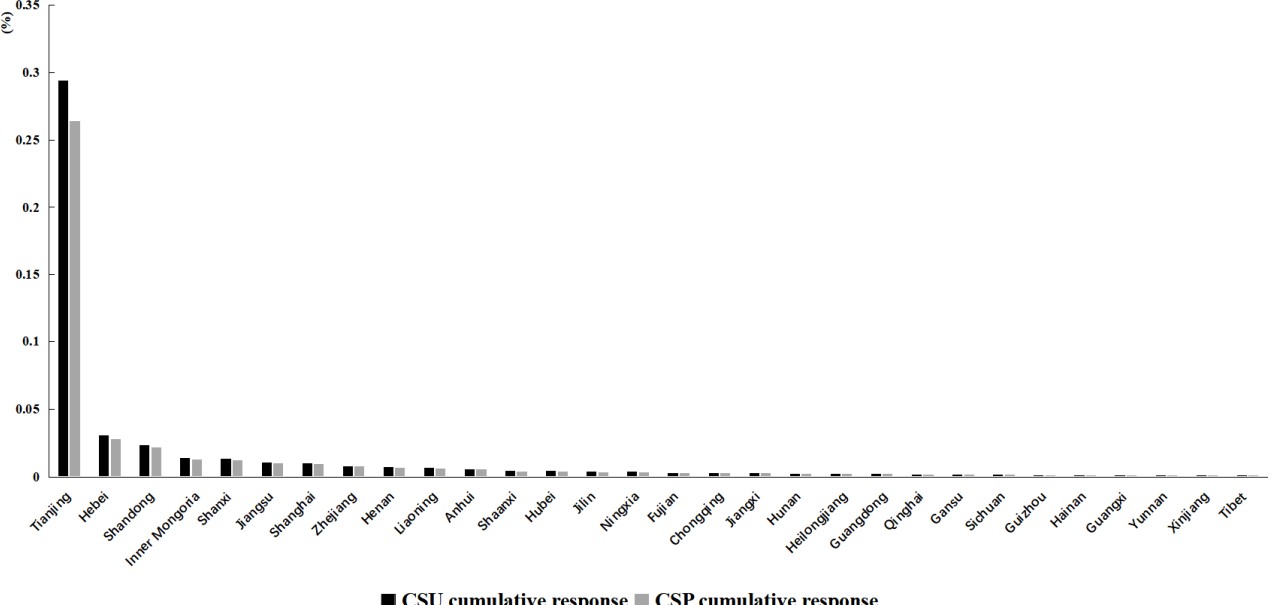

**Figure 7.** Cumulative impact of CSU shocks in other provinces on Beijing.

Overall, the impact of positive unit standard deviation shocks on CSP and CSU in other provinces was significantly smaller than in Beijing. This is primarily due to Beijing's central role as a natural political, economic, and consumption center in the entire

country. Beijing acts as a critical "bridge" and "link" in the spatial linkage of consumption with neighbors and exerts strong control over the goods and services in circulation. In addition, Beijing's higher income and consumption levels, diverse consumption content, and advanced consumption methods and concepts make it a model for residents of other provinces to compete with and emulate. It is worth noting that Beijing's control over the circulation of goods and services is also an important factor contributing to its dominant position in this regard.

To compare the magnitude of the two-way shocks, this study examines the examples of Tianjin and Hebei, two other members of the Beijing–Tianjin–Hebei integration. The Tianjin CSP can drive cumulative growth of 0.266% and 0.226% for the Beijing CSP and CSU, respectively. Conversely, Beijing's CSP and CSU can grow cumulatively by 0.322% and 0.315%, respectively, in response to the Tianjin CSU shock. This impact is smaller than the cumulative effect of the Beijing CSP and CSU shocks on Tianjin. Two primary factors account for this phenomenon. First, Tianjin lags behind Beijing in terms of economic strength, resident income, and consumption levels. For example, Tianjin's per capita disposable income in 2020 was only 63.16% of Beijing's, at 43,854,000 yuan. Second, Tianjin's role was to support Beijing in undertaking some of its economic and external transport functions, making it a satellite city of Beijing. Thus, its "supporting" role determines that its impact on Beijing is weaker than the reverse impact. Although Hebei is part of the same economic region as Beijing and Tianjin, it is positioned to take over the transfer of industries from these two cities. As a passive recipient, it was often difficult for Hebei to create reverse exports. Furthermore, Hebei's development has many shortcomings, such as a per capita GDP below the national average, less than half that of Beijing and Tianjin, and a poverty belt around Beijing and Tianjin. Finally, the Hebei CSP and CSU units' positive standard deviation shock brings a cumulative growth of both the Beijing CSP and CSU of less than 0.04%.

## 6. Conclusions

The main findings of this study are as follows: First, the CSP and CSU of each province in China witnessed rapid growth during the observation period, displaying a "stepped" development pattern with decreasing values from east to west. However, the growth rate of the western and central provinces was higher than that of the lagging provinces, indicating a "catch-up effect" of the relatively lagging provinces on the high-development provinces. Second, there is a significant positive spatial correlation between CSP and CSU with co-existing diffusion and agglomeration. The pattern of agglomeration is characterized by "high–high" and "low–low" agglomeration, with fewer "high–high" clustering provinces mainly located in the eastern coastal and central fast-rising regions. Third, the overall spatial divergence between CSP and CSU is decreasing, and inter-regional differences constitute the primary cause of spatial differentiation, followed by intra-regional differentiation. Fourth, the CSP and CSU have a positive feedback mechanism that promotes each other, and there is a "center–periphery" spatial spillover effect. The spillover effect of the "center" is proportional to the inter-provincial geographical distance and economic closeness. The "center" is also subject to relatively weak reverse shocks from the "periphery", with the reverse shock effect limited by economic ties and geographical distance.

Several policy insights are derived based on these findings. First, during the sample observation period, provinces with higher levels of CSP and CSU exhibited more developed economies, suggesting that enhancing the development of CSP and CSU can effectively boost the economic growth of provinces, particularly less-developed ones. To achieve this, government support should be combined with market mechanisms, the income distribution system should be improved, people's livelihoods should be addressed, a conducive consumption environment should be created, and the population's consumption potential should be harnessed through various means. Additionally, consumer protection should be continuously strengthened, the consumption capacity should be increased, and the demand for consumption upgrading should be met. Second, although China's CSP and

CSU have achieved relatively rapid development in the long term, significant disparities in the development starting points of different provinces exist, leading to serious spatial agglomeration and unevenness. Thus, it is necessary to enhance CSP and CSU at the macro level, accurately position provinces in different regions, accelerate the improvement of effective inter-regional exchange and cooperation mechanisms, promote inter-regional economic and trade cooperation and cross-regional consumption, and reduce inter-regional differences in CSP and CSU. Third, the spatial spillover between CSPs and CSUs has formed a mutual promotion mechanism characterized by a "center–periphery" spatial spillover, with the spillover effect of consumption centers helping to promote coordinated interaction between CSPs and CSUs in the region. Therefore, based on a comprehensive evaluation of the comparative advantages of provinces and cities in terms of their location, radiation capacity, and consumption levels, new highlands for consumption should be built at multiple points across the country. This will strengthen the role of international or regional consumption centers, such as Beijing, in terms of agglomeration, radiation, and leadership and enhance the mutual promotion between the "center" and the "periphery" in terms of consumption. This will facilitate the formation of a synergistic and coupled development pattern to promote high-quality economic development on a larger scale.

**Author Contributions:** Conceptualization, X.X. and A.Y.; methodology, X.X. and A.Y.; software, X.X.; validation, X.X.; formal analysis, X.X.; investigation, X.X.; resources, X.X. and A.Y.; data curation, X.X.; writing—original draft preparation, X.X.; writing—review and editing, X.X.; visualization, X.X.; supervision, A.Y.; project administration, A.Y.; funding acquisition, A.Y. All authors have read and agreed to the published version of the manuscript.

**Funding:** This study was supported by the National Natural Science Foundation of China (No. 71171057), the National Natural Science Foundation of China (No. 71571046), and the National Natural Science Foundation of China (No. 72073030).

**Data Availability Statement:** The datasets used during the current study are available from the corresponding author upon reasonable request.

**Conflicts of Interest:** The authors declare no conflict of interest.

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
