# Peer review of "Spatial-Temporal Differentiation and Spatial Interaction Effect Analysis of Residents’ Consumption Capacity and Consumption Upgrading in China"

_systems, doi:10.3390/systems11050219_

Round 1

Reviewer 1 Report

The formulas around equation 1 may contain errors, and there is a mix of capitalized Y and small y. In S², a square may be missing.  The mentioned "Thayer index" should be explained.

Figure 1 does not show what is written below the graph. It shows a time-series plot of CSP by region.

Decimals in the numbers below Figure 1 should be reasonable, maybe one or two, not excessive.

There is no "constant decline". It may be a persistent decline.

In section 5.2, the statistical terminology is incorrect. Alternative hypotheses are not rejected. The null of "no Granger causality" is rejected at 1%. Alternatives are not accepted either. In table 4, the text "reasons why CSP is CSU's Granger" does not make sense.

Impulse response graphs in Figure 3 could be improved by introducing significance, for example via confidence intervals.

An issue with this paper is its language. I surmise that some of my problems in following the logical flow are based on unusual words. For example, a "consumption agent" may simply be a consumer. The wording is not wrong but irritating. By contrast, "consumption capacity promoting consumption upgrading" is entirely unclear, it sounds like an arbitrary cluster of words.

Otherwise, grammar is problematic. There are instances where an article "the" should be dropped, and others where two sentences should be fused into one (the first sentence of section 4.1, for example).

Beijing is clearly an important city. The wording "international consumption centre city" makes little sense however.

Author Response

Point 1: The formulas around equation 1 may contain errors, and there is a mix of capitalized Y and smally. In S², a square may be missing.  The mentioned "Thayer index" should be explained.

Response 1: This revision given by the reviewer was accepted. Thank you for reviewing our translations. We regret that some problems have arisen in the translation of the article and we have carefully checked the translation and made the following corrections.

(1) By re-finding the corresponding literature, we re-checked all the formulas covered in the article and revised them for possible errors.

(2) We apologise for the error in the translation of the concept "Theil index". We have replaced "Thayer index" with "Theil index" as it should be in the text. Theil index is first proposed by Theil and Henri in 1967, it is widely used to measure differences in levels of economic development between regions, but it suffers from the difficulty of addressing the problem of crossover between samples between regions.

Please check our translation again and feel free to contact us if you have any further questions or suggestions.

Point 2: Figure 1 does not show what is written below the graph. It shows a time-series plot of CSP by region.Decimals in the numbers below Figure 1 should be reasonable, maybe one or two, not excessive.There is no "constant decline". It may be a persistent decline.

Response 2: This revision given by the reviewer was accepted. After comparing the original program output with the Chinese text of the article, we found that the vertical axis was incorrectly set in the plotting of the CSU's Dagum Gini coefficient and decomposition results; at the same time, we also found that some of the content was translated incorrectly during the translation process. For this reason, we have revised and replaced the images in this section based on the original results and the Chinese writing to ensure that the relevant content descriptions are consistent with those presented in Figure 1 and t Figure 2. To ensure the objectivity of the results, we further provide detailed Dagum Gini coefficients and decomposition results for CSU and CSP (with three decimal places reserved), as shown in Tables 1 and 2.

Table1 The Dagum Gini coefficient and decomposition results of CSP

Year

Whole

Intra-regional Gini coefficient

Inter-regional Gini coefficient

Contribution (%)

East

Central

West

Central-East

West-East

West-Central

intra-regional

inter-regional

Super-variable density

2008

0.220

0.184

0.048

0.103

0.239

0.348

0.144

23.004

75.988

1.007

2009

0.215

0.172

0.052

0.108

0.236

0.343

0.138

22.592

77.162

0.246

2010

0.180

0.145

0.055

0.082

0.209

0.287

0.106

22.248

77.088

0.664

2011

0.193

0.147

0.057

0.105

0.215

0.307

0.124

22.295

77.120

0.585

2012

0.180

0.133

0.063

0.105

0.199

0.284

0.122

22.333

75.536

2.131

2013

0.166

0.131

0.052

0.091

0.184

0.263

0.109

22.668

75.251

2.081

2014

0.166

0.125

0.064

0.092

0.183

0.262

0.114

22.413

75.241

2.346

2015

0.157

0.121

0.068

0.080

0.178

0.246

0.105

22.542

74.464

2.994

2016

0.156

0.126

0.070

0.069

0.182

0.244

0.099

22.644

74.104

3.252

2017

0.159

0.126

0.070

0.072

0.187

0.249

0.100

22.529

74.340

3.131

2018

0.160

0.127

0.074

0.086

0.180

0.246

0.107

23.494

73.008

3.498

2019

0.159

0.132

0.068

0.084

0.178

0.243

0.105

23.933

72.603

3.463

2020

0.151

0.130

0.054

0.075

0.171

0.235

0.097

23.625

73.919

2.455

Average

0.174

0.138

0.061

0.089

0.196

0.274

0.113

22.794

75.063

2.143

Table2 The Dagum Gini coefficient and decomposition results of CSU

YEAR

Whole

Intra-regional Gini coefficient

Inter-regional Gini coefficient

Contribution (%)

East

Central

West

Central-East

West-East

West-Central

intra-regional

inter-regional

Super-variable density

2008

0.251

0.221

0.066

0.129

0.308

0.372

0.115

25.170

74.830

0.000

2009

0.234

0.227

0.046

0.117

0.294

0.341

0.096

26.515

73.485

0.000

2010

0.234

0.220

0.042

0.123

0.296

0.343

0.099

26.039

73.961

0.000

2011

0.229

0.207

0.044

0.126

0.292

0.335

0.100

25.663

74.295

0.042

2012

0.223

0.204

0.051

0.125

0.284

0.324

0.101

25.937

72.847

1.216

2013

0.214

0.193

0.057

0.112

0.275

0.314

0.094

25.457

73.923

0.621

2014

0.204

0.189

0.060

0.113

0.258

0.296

0.096

26.137

72.193

1.670

2015

0.199

0.191

0.054

0.105

0.255

0.290

0.088

26.402

72.205

1.393

2016

0.202

0.192

0.054

0.104

0.260

0.296

0.088

26.085

72.455

1.460

2017

0.200

0.188

0.056

0.099

0.259

0.294

0.087

25.736

72.987

1.277

2018

0.199

0.189

0.068

0.093

0.254

0.293

0.090

25.944

72.743

1.312

2019

0.204

0.187

0.068

0.094

0.257

0.303

0.093

25.348

73.799

0.853

2020

0.213

0.193

0.065

0.103

0.267

0.317

0.100

25.219

73.180

1.602

Average

0.216

0.200

0.056

0.111

0.274

0.317

0.096

25.819

73.300

0.880

Point 3: In section 5.2, the statistical terminology is incorrect. Alternative hypotheses are not rejected. The null of "no Granger causality" is rejected at 1%. Alternatives are not accepted either. In table 4, the text "reasons why CSP is CSU's Granger" does not make sense.

Response 3: This revision given by the reviewer was accepted.The section has been reworked to ensure that the statistical terminology is correct, with reference to the original data output from the original software and the Chinese writing.

Point 4: Impulse response graphs in Figure 3 could be improved by introducing significance, for example via confidence intervals.

Response 4: Since Beenstock and Felsenstein proposed the spatial vector autoregressive model in 2008, the SPVAR model has been used in practice, but as the model is a frontier econometric model, there are still many areas that need to be analysed and studied in depth, among which the software implementation and output is one of the issues that need to be addressed. The current Matlab-based SPVAR model can output impulse response values, but it is still unable to output confidence intervals for impulse response values in the same way as the traditional VAR. In the future, we will further strengthen this area of research in order to obtain more comprehensive and accurate SPVAR model estimation results. Some of the outputs are shown in Table 3:

Table 3 Example output of the corresponding estimates of the SPVAR model impulses

N.

Beijing

Tianjin

Hebei

Shanxi

Neimenggu

Liaoning

Jilin

Heilongjiang

Shanghai

Jiangsu

1

1.14745

0.00000

0.00000

0.00000

0.00000

0.00000

0.00000

0.00000

0.00000

0.00000

2

-0.19971

0.15570

0.06004

0.03039

0.04851

0.03171

0.01563

0.01448

0.00364

0.00459

3

0.12169

0.03531

0.02102

0.01466

0.01736

0.01335

0.00901

0.00806

0.00228

0.00284

4

0.00646

0.02772

0.01861

0.01398

0.01559

0.01243

0.00915

0.00830

0.00249

0.00305

5

0.01855

0.01420

0.00951

0.00749

0.00818

0.00664

0.00542

0.00512

0.00173

0.00204

6

0.00639

0.00760

0.00598

0.00500

0.00521

0.00442

0.00377

0.00356

0.00130

0.00151

7

0.00449

0.00433

0.00330

0.00280

0.00293

0.00252

0.00225

0.00217

0.00088

0.00099

8

0.00224

0.00233

0.00193

0.00169

0.00173

0.00153

0.00140

0.00135

0.00059

0.00066

9

0.00131

0.00132

0.00108

0.00096

0.00098

0.00089

0.00083

0.00081

0.00038

0.00042

10

0.00072

0.00073

0.00062

0.00056

0.00057

0.00052

0.00049

0.00048

0.00024

0.00026

11

0.00041

0.00041

0.00035

0.00032

0.00033

0.00030

0.00029

0.00028

0.00015

0.00016

12

0.00023

0.00023

0.00020

0.00018

0.00019

0.00017

0.00017

0.00017

0.00010

0.00010

13

0.00013

0.00013

0.00011

0.00011

0.00011

0.00010

0.00010

0.00010

0.00006

0.00006

Unit: %

Point 5: Comments on the Quality of English Language. An issue with this paper is its language. I surmise that some of my problems in following the logical flow are based on unusual words. For example, a "consumption agent" may simply be a consumer. The wording is not wrong but irritating. By contrast, "consumption capacity promoting consumption upgrading" is entirely unclear, it sounds like an arbitrary cluster of words. Otherwise, grammar is problematic. There are instances where an article "the" should be dropped, and others where two sentences should be fused into one (the first sentence of section 4.1, for example).Beijing is clearly an important city. The wording "international consumption centre city" makes little sense however.

Response 5: This revision given by the reviewer was accepted. Thank you for reviewing our article and providing feedback on the language. We have made revisions to the article as per your request and hope that these changes will make the article more clear and understandable. We appreciate your valuable input and will continue to strive for improving the quality of our articles.

Reviewer 2 Report

The manuscript is well-written and interesting. I can find only minor issues that need to be addressed in order to make the manuscript suitable for publication. Please find my remarks below:

1. In tables 1 and 2 I recommend to make current lines thicker and add more thin lines to separate the groups of indicators (Resident income, social security, etc.). Now we can only guess, which specific indicator belongs to which group.

2. In table 3 I think that the columns p-value are superfluous, when the asterisks presenting the significance levels, are provided.

3. There is something wrong with the description of subscripts i and t under the equation (3).

4. Please put more attention in the notation of references and citations. Now they are not in line with the formal requirements.

English is mostly fine, I just noticed some missing spaces.

Author Response

Response to Reviewer 2 Comments

Point 1: In tables 1 and 2 I recommend to make current lines thicker and add more thin lines to separate the groups of indicators (Resident income, social security, etc.). Now we can only guess, which specific indicator belongs to which group.

Response 1: This revision given by the reviewer was accepted. With reference to the journal's submission instructions and the published literature, we have reformatted Forms 1 and 2 to separate the groups of indicators by setting up table lines, etc.

Point 2: In table 3 I think that the columns p-value are superfluous, when the asterisks presenting the significance levels, are provided.

Response 2: This revision given by the reviewer was accepted. Considering that the z-values corresponding to the Moran index are already provided in Table 3 and have been identified at what level they are significant, the p-values corresponding to the Moran values will be removed from this paper.

Point 3: There is something wrong with the description of subscripts i and t under the equation (3).

Response 3: This revision given by the reviewer was accepted. By re-finding the corresponding literature, we re-checked all the formulas covered in the article and revised them for possible errors.

Point 4: Please put more attention in the notation of references and citations. Now they are not in line with the formal requirements.

Response 4: This revision given by the reviewer was accepted. With reference to the journal's submission instructions and published literature, we have revised the notation of references and citations in the article to ensure that it can better meet the journal's requirements.
